# Fatigue in incident peritoneal dialysis and mortality: A real-world side-by-side study in Brazil and the United States

Murilo Guedes[1‡], Liz Wallim[1‡], Camila R. Guetter[2], Yue Jiao[3], Vladimir Rigodon[1,4], Chance Mysayphonh[1,4], Len A. Usvyat[3], Pasqual Barretti[5], Peter Kotanko[6,7], John W. Larkin[3], Franklin W. Maddux[8], Roberto Pecoits-Filho[1,9], Thyago Proenca de Moraes[1]*

1 Pontifícia Universidade Católica do Paraná, Curitiba, PR, Brazil, 2 Johns Hopkins Bloomberg School of Public Health, Baltimore, MD, United States of America, 3 Fresenius Medical Care Global Medical Office, Waltham, MA, United States of America, 4 Fresenius Medical Care North America, Waltham, MA, United States of America, 5 Universidade do Estado de São Paulo, Botucatu, SP, Brazil, 6 Renal Research Institute, New York, NY, United States of America, 7 Icahn School of Medicine at Mount Sinai, New York, NY, United States of America, 8 Fresenius Medical Care AG & Co. KGaA, Global Medical Office, Bad Homburg, Germany, 9 Arbor Research Collaborative for Health, Ann Arbor, Michigan, United States of America

‡ MG and LW are contributed equally to this work and share first authorship.
* thyago.moraes@pucpr.br

**Data Availability Statement:** All relevant data are within the manuscript to draw results and conclusions. Raw data on cohort including patients excluded from this analysis are available upon

## Abstract

### Background

We tested if fatigue in incident Peritoneal Dialysis associated with an increased risk for mortality, independently from main confounders.

### Methods

We conducted a side-by-side study from two of incident PD patients in Brazil and the United States. We used the same code to independently analyze data in both countries during 2004 to 2011. We included data from adults who completed KDQOL-SF vitality subscale within 90 days after starting PD. Vitality score was categorized in four groups: >50 (high vitality), ≥40 to ≤50 (moderate vitality), >35 to <40 (moderate fatigue), ≤35 (high fatigue; reference group). In each country's cohort, we built four distinct models to estimate the associations between vitality (exposure) and all-cause mortality (outcome): (i) Cox regression model; (ii) competing risk model accounting for technique failure events; (iii) multilevel survival model of clinic-level clusters; (iv) multivariate regression model with smoothing splines treating vitality as a continuous measure. Analyses were adjusted for age, comorbidities, PD modality, hemoglobin, and albumin. A mixed-effects meta-analysis was used to pool hazard ratios (HRs) from both cohorts to model mortality risk for each 10-unit increase in vitality.

### Results

We used data from 4,285 PD patients (Brazil n = 1,388 and United States n = 2,897). Model estimates showed lower vitality levels within 90 days of starting PD were associated with a

reasonable written requests to the institutions Pontifícia Universidade Católica do Paraná, School of Medicine, Rua Imaculada Conceição 1155, Curitiba, PR, 80215-182, Brazil and Fresenius Medical Care, Global Medical Office, 920 Winter St., Waltham, MA, 02451. Requests can also be sent to the authors Thyago Proenca de Moraes (email: thyago.moraes@pucpr.br) and/or John Larkin (email: john.larkin@freseniusmedicalcare.com).

**Funding:** The funders for the BRAZPD II study (Baxter Healthcare and Pontifícia Universidade Católica do Paraná) did not have any additional role in the study design, data collection and analysis, decision to publish, or preparation of the manuscript, outside the specific roles of authors in the author contributions. The funders for the United States analysis (Fresenius Medical Care) provided support in the form of salaries for authors (YJ, VMR, CM, LAU, PK, JWL, FWM), data, infrastructure and approval to conduct project, but did not have any additional role in the study design, data collection and analysis, decision to publish, or preparation of the manuscript, outside the specific roles of authors in the author contributions.

**Competing interests:** I have read the journal's policy and the authors of this manuscript have the following competing interests: MG, LW, CRG, VMR, CM are students at Pontifícia Universidade Católica do Paraná. CRG is a student at Johns Hopkins Bloomberg School of Public Health. VMR, CM, YJ, JWL, LAU, FWM are employees of Fresenius Medical Care. PK is an employee of Renal Research Institute, a wholly owned subsidiary of Fresenius Medical Care. LAU, PK, FWM have share options/ownership in Fresenius Medical Care. JWL, LAU, PK, FWM are an inventor on patent(s) in the field of dialysis. PK receives honorarium from Up-To-Date and is on the Editorial Board of Blood Purification and Kidney and Blood Pressure Research. FWM has directorships in Fresenius Medical Care Management Board, Goldfinch Bio, and Vifor Fresenius Medical Care Renal Pharma. RPF, TPM are employed by Pontifícia Universidade Católica do Paraná, and are recipients of scholarships from the Brazilian Council for Research (CNPq). RPF is employed by Arbor Research Collaborative for Health, and receives research grants, consulting fees, and honoraria from AstraZeneca, Novo Nordisc, Akebia Therapeutics, and Fresenius Medical Care. JWL, PB, TPM are guest editors on the Editorial Board of Frontiers in Physiology. This does not alter our adherence to PLOS ONE policies on sharing data and materials.

higher risk of mortality, which was consistent in Brazil and the United States cohorts. In the multivariate survival model, each 10-unit increase in vitality score was associated with lower risk of all-cause mortality in both cohorts (Brazil HR = 0.79 [95%CI 0.70 to 0.90] and United States HR = 0.90 [95%CI 0.88 to 0.93], pooled HR = 0.86 [95%CI 0.75 to 0.98]). Results for all models provided consistent effect estimates.

## Conclusions

Among patients in Brazil and the United States, lower vitality score in the initial months of PD was independently associated with all-cause mortality.

## Introduction

People with kidney failure commonly suffer from reduced health-related quality of life (HRQOL). Nephrology care and research paradigms are in a crescent shift towards more person-centered models. There are notable initiatives underway at scientific and organizational levels promoting active patient participation and engagement in kidney failure care [1–3]. In this sense, life participation and vitality have been shown to be valued outcomes for kidney failure patients, caregivers, and providers alike [2, 4–6].

Fatigue is a multi-dimensional phenomenon characterized by weakness, tiredness, and physical and/or mental exhaustion [7]. Although fatigue is a natural adaptive response to stressors signaling the body to rest, prolonged fatigue can be disabling. In patients with kidney failure, fatigue is multifactorial with an interplay between physiological, sociodemographic, psychological, and dialysis-related factors [7]. Particularly in patients with CKD, fatigue may largely be the consequence of an allostatic inflammatory state that is commonly observed through chronically elevated levels of pro-inflammatory cytokines (e.g. interleukin-6 (IL-6) and tumor necrosis factor-alpha (TNF-$\alpha$)) [8]. Among individuals with kidney failure undergoing hemodialysis, lower physical health-related quality of life and, particularly, vitality has been consistently associated with worse clinical outcomes, including all-cause mortality [9, 10]. However, patients undergoing peritoneal dialysis (PD) are different compared to those in HD, for several reasons [11]. First, they tend to be younger and more functional. Second, the distribution of comorbidities is different between modalities. Therefore, fatigue causes, and associated factors, are often different between HD and PD. Finally, the transition period to dialysis initiation requires more active patient engagement in PD. Fatigue, then, can disproportionally pose higher risk for adverse outcomes in PD patients.

In the present study, we aimed to conduct a parallel analysis of two nationally representative cohorts of PD patients in Brazil and the United States to test the hypothesis that higher fatigue levels would associate with higher risks for all-cause mortality, even after adjustment for common and clinically important confounders.

## Materials and methods

### Description of cohorts

We conducted side-by-side analyses of real-world data in two cohorts of adult PD patients in Brazil and the United States from December 2004 to January 2011.

In Brazil, we used observational data that were collected in BRAZPD II in a nationwide cohort of PD patients across 122 centers [12]. The BRAZPD II observational cohort data were

captured prospectively in a de-identified manner under a protocol approved by the Pontifícia Universidade Católica do Paraná Ethics Review Board (Curitiba, Parana, Brazil; approval number 25000.187284/2004-1), and all patients provided informed consent.

In the United States, using data collected during the provision of routine medical care among PD patients treated in the national network of dialysis centers (Fresenius Kidney Care, Waltham, MA, United States) of an integrated kidney disease healthcare company (Fresenius Medical Care North America, Waltham, MA, United States). This analysis of de-identified secondary data was conducted under a protocol approved by New England Independent Review Board, which provided a waiver of informed consent per title 45 of the United States Code of Federal Regulations part 46.116(f) (Needham Heights, MA, United States; NEIRB# 17-1334030-1).

The parallel analyses performed for this study were conducted in accordance with the Declaration of Helsinki.

### Patient eligibility

In both cohorts, we included data from all adult kidney failure patients (age $\geq$18 years) who completed the KDQOL-SF v1.3 survey within 90 days of starting continuous ambulatory peritoneal dialysis (CAPD) or continuous cycling peritoneal dialysis (CCPD) for kidney replacement therapy and remained on PD for at least 90 days. Included patients were required to have performed self-care, or caregiver-assisted, PD in a home setting. We excluded data from individuals who died or had a technique failure event within 90 days after initiating PD. Also, we excluded data from subjects treated outside training with in-center PD and/or any clinician-assisted PD.

### Exposure definition

Fatigue was measured by the KDQOL-SF v1.3 survey using the vitality subscale [13]. Fatigue is defined from the vitality subscale by an average of the answers to the following KDQOL-SF questions asking "how much of the time during the past 4 weeks": 9a. "Did you feel full of pep?", 9e. "Did you have a lot of energy?", 9g. "Did you feel worn out?" 9i. "Did you feel tired?". All answers have six levels structured as a Likert scale, and for each response, a score varying from 0 to 100 is attributed. The average over the question scores is the vitality score. Therefore, lower scores represent higher fatigue. In this study, we categorized vitality/fatigue levels as >50 (high vitality), $\geq$40 to $\leq$50 (moderate vitality), >35 to <40 (moderate fatigue), $\leq$35 (high fatigue). The categorization thresholds for vitality/fatigue levels were selected considering cutoffs used in prior studies [14, 15] and distributions seen in the cohorts evaluated.

Baseline demographic/clinical characteristics were captured from data within 90 days of starting PD in both cohorts. Residual kidney function (RKF) was categorized from baseline values as a binary variable below or above/equal to 100 mL of urine volume per 24 hours.

### Outcome definition

The primary outcome of this study was all-cause mortality. Time at risk started at the end of 90 days from initiation of PD. Patients were censored if they were transferred out of the providers' dialysis centers or had a PD technique failure event in the follow-up period, defined as a switch to an HD modality for >90 days.

### Statistical analysis

We summarized data across vitality categories. Categorical variables were calculated number and proportions, and continuous variables were computed as the mean and standard deviation

(SD). We built Kaplan-Meier curves treating vitality categories as the exposure variable. To estimate the association between fatigue and all-cause mortality, we used four distinct models to address different underlying assumptions for data modeling and test robustness of the estimates. The first model consisted of a Cox proportional hazard model. The second model was comprised of a competing risk analysis using the Fine and Grey method, which defined technique failure as a competing risk for the all-cause mortality outcome. The third model was a multilevel survival analysis, considering clinic-level values as a random variable in a mixed model. For the first three models, the effect estimates considered the high fatigue category (vitality score ≤35) as the reference exposure variable. We assessed the proportional hazards assumption by inspecting Kaplan-Meier plots and testing the interaction between log time and vitality categories. Finally, in the fourth model, we constructed a multivariate regression model with smoothing splines treating vitality as a continuous measure. A mixed-effects meta-analysis was used to pool hazard ratios (HRs) from both cohorts to model mortality risk for each 10-unit increase in vitality.

All analyses were adjusted by the following set of confounders: a history of diabetes, coronary artery disease, peripheral artery disease, previous history of HD treatment, initial PD modality (CAPD/CCPD), hemoglobin, creatinine, and age. These confounders were selected based on a priori assumptions using expert insights about the potential causal structure underlying the data. Assumptions (e.g., predicted correlations) were checked in the data distributions and generally followed previous data analyses in the cohort [16].

The unique datasets on cohorts from Brazil and the United States were structured in a universal format, and each independent analysis was performed in parallel using a single source of code using R software version 4.0.0.

## Results

In Brazil, of a total of 9,905 PD patients in the BRAZPD cohort, we included data from 1,388 incident PD patients who had a vitality score at baseline that was collected within 90 days of PD initiation. In the United States, of a total of 16,141 PD patients at the provider's national network, we included data from 2,897 incident patients who had a vitality score at baseline that was collected within 90 days of PD initiation (**Fig 1**).

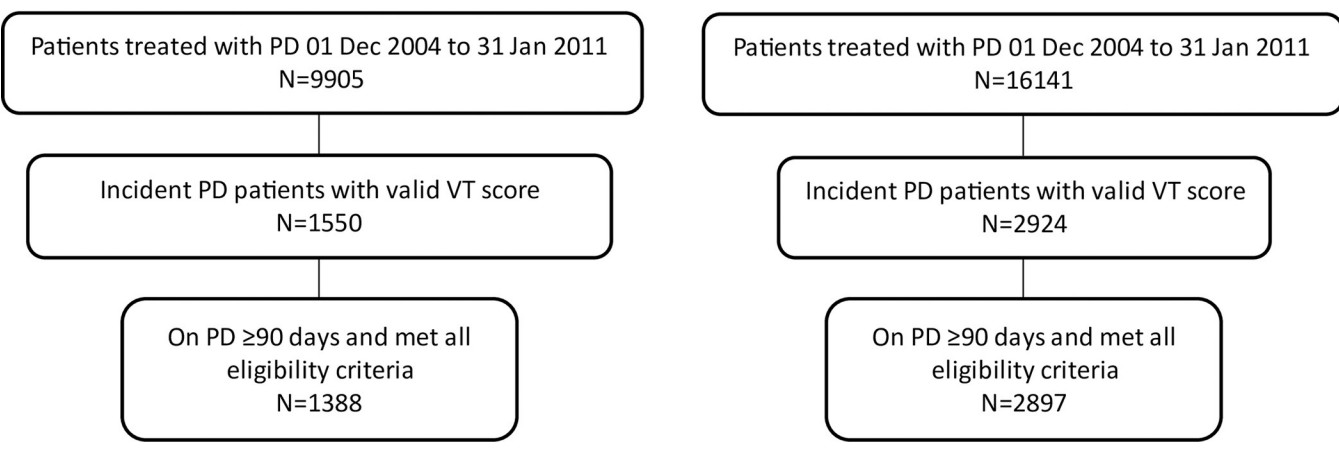

**Fig 1. Flow chart of patients included in the Brazil and United States cohorts.**

Table 1. Patient characteristics in the Brazil and United States cohorts.

| Cohort | Brazil Cohort | | | | | United States Cohort | | | | |
|---|---|---|---|---|---|---|---|---|---|---|
| Vitality Score Group | Overall | <35 | 35–40 | 40–50 | >50 | Overall | <35 | 35–40 | 40–50 | >50 |
| n | 1388 | 347 | 212 | 535 | 294 | 2897 | 820 | 194 | 435 | 1448 |
| Age (years) | 58.5 (15.5) | 61.5 (14.6) | 59.6 (15.7) | 58.1 (15.9) | 54.8 (14.7) | 55.8 (15.3) | 56.9 (15.0) | 57.3 (14.8) | 57.0 (15.2) | 54.6 (15.5) |
| Sex (Male) | 663 (47.8) | 157 (45.2) | 99 (46.7) | 265 (49.5) | 142 (48.3) | 1578 (54.5) | 429 (52.3) | 110 (56.7) | 247 (56.8) | 792 (54.7) |
| Race (white) | 827 (59.6) | 203 (58.5) | 122 (57.5) | 323 (60.4) | 179 (60.9) | 2050 (71.6) | 642 (79.2) | 144 (75.8) | 305 (70.9) | 959 (67.0) |
| BMI (kg/m2) | 24.6 (4.6) | 24.7 (4.8) | 24.4 (4.5) | 24.7 (4.5) | 24.7 (4.8) | 28.9 (9.2) | 29.2 (7.8) | 28.1 (6.1) | 27.9 (6.4) | 29.2 (10.9) |
| DM | 598 (43.1) | 175 (50.4) | 97 (45.8) | 212 (39.6) | 114 (38.8) | 891 (30.8) | 281 (34.3) | 52 (26.8) | 138 (31.7) | 420 (29.0) |
| CAD | 279 (20.1) | 79 (22.8) | 42 (19.8) | 107 (20.0) | 51 (17.3) | 349 (12.0) | 120 (14.6) | 24 (12.4) | 64 (14.7) | 141 (9.7) |
| PAD | 314 (22.6) | 88 (25.4) | 41 (19.3) | 121 (22.6) | 64 (21.8) | 180 (6.2) | 45 (5.5) | 15 (7.7) | 37 (8.5) | 83 (5.7) |
| Previous HD | 592 (42.7) | 147 (42.4) | 90 (42.5) | 218 (40.7) | 137 (46.6) | 1265 (43.7) | 339 (41.3) | 89 (45.9) | 201 (46.2) | 636 (43.9) |
| Total Exchange Volume | 11.2 (3.9) | 11.4 (4.1) | 11.1 (4.4) | 10.7 (3.4) | 12.1 (4.1) | 9.60 (2.08) | 9.65 (2.04) | 9.79 (2.51) | 9.45 (1.92) | 9.60 (2.11) |
| Dialysis vintage (months) | 28.1 (30.6) | 23.7 (24.2) | 28.8 (32.0) | 27.6 (30.6) | 33.6 (35.3) | 8.2 (19.2) | 8.1 (21.1) | 9.4 (19.5) | 9.1 (19.6) | 7.8 (17.9) |
| RKF | 888 (64.0) | 218 (62.8) | 129 (60.8) | 352 (65.8) | 189 (64.3) | 1755 (97.7) | 484 (97.8) | 113 (97.4) | 256 (98.1) | 902 (97.6) |
| Creatinine (mg/dL) | 6.8 (3.1) | 6.5 (3.1) | 6.4 (2.8) | 7.0 (3.1) | 7.2 (3.1) | 7.8 (3.8) | 7.4 (3.6) | 7.7 (4.3) | 7.8 (3.6) | 8.0 (3.9) |
| Hemoglobin (mg/dL) | 10.8 (1.7) | 10.8 (1.8) | 10.8 (1.7) | 11.0 (1.7) | 10.6 (1.8) | 12.3 (1.5) | 12.2 (1.4) | 12.4 (1.5) | 12.3 (1.5) | 12.4 (1.5) |

Mean (SD) or count (%). DM: Diabetes Mellitus. CAD: Coronary Artery Disease. PAD: Peripheral Artery Disease. HD: Hemodialysis. BMI: Body mass index. RKF: Residual kidney function.

In general, patient characteristics were similar between cohorts, with a slightly higher prevalence of cardiovascular diseases in the cohort for Brazil, as well as longer dialysis vintage, and lower RKF as compared to the United States (Table 1). In both cohorts, patients with lower vitality scores tended to have a higher prevalence of comorbidities, lower proportion with RKF (≥100 mL urine output per 24 hours), and shorter vintage. The median follow-up period was 14 months (interquartile range (IQR): 7 to 26) for patients in Brazil and 21 months (IQR: 11 to 39) for patients in the United States.

In the univariate survival analysis of PD patients in Brazil, patients who were older, those with higher prescribed total exchange volume per 24 hours, history of coronary artery disease, and diabetes had a lower probability of survival over the follow-up period (S1 Table). The presence of RKF and longer dialysis vintage were associated with longer survival. Consistently, in the parallel univariate survival analysis of PD patients in the United States, patients who were older, white, had higher prescribed total exchange volume per 24 hours, history of coronary artery disease, diabetes, and peripheral artery disease had a lower likelihood of survival (S1 Table). RKF, higher creatinine, and higher hemoglobin were associated with a longer survival. In contrast to findings among patients in Brazil, patients in the United States who had longer dialysis vintage exhibited a shorter survival. Kaplan-Meier curves for both cohorts are depicted in (S1 and S2 Figs). Consistent in both cohorts, patients reporting higher vitality score categories survived longer during the follow-up period.

In PD patients in Brazil, the mortality incidence rates per 100 patients-years (p100py) were 16.2, 11.9, 11.5 and 7.7 from the lowest to the highest vitality category. In the United States the mortality incidence rates p100py were 17.5, 17.7, 15.0, 12.1 from the lowest to highest vitality category. The adjusted hazard ratios (HRs) by the vitality category for the three models are plotted in Fig 2 for both cohorts. Consistently across models and cohorts, patients in the high vitality group (vitality score >50) had a lower risk of mortality; the effect estimates ranged from a 61% to 32% lower risk of mortality in Brazil and 34% to 22% lower risk of mortality in the United States (Table 2)

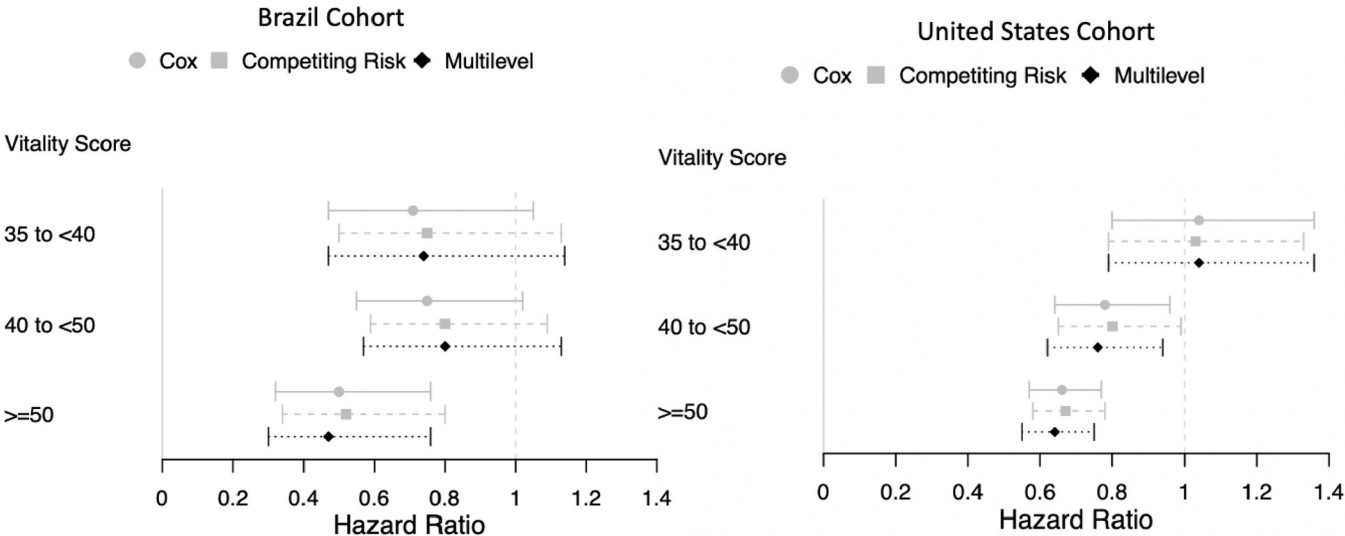

**Fig 2. Forrest Plot of model estimates in the cohorts from Brazil and the United States.** Models are adjusted for history of diabetes, coronary artery disease, peripheral artery disease, previous history of HD treatment, initial PD modality, hemoglobin, creatinine, and age.

In the multivariate regression model that treated vitality scores as a continuous variable, we found the vitality score had an inverse linear association with mortality in both cohorts; the HR for death per 10-point increase in vitality score was 0.79 [95% confidence interval (CI) 0.70 to 0.90] for patients in Brazil and 0.90 [95%CI 0.88 to 0.93] for patients in the United States. Smoothing spline regression plots are shown in **Fig 3**. The pooled HR for each 10-unit increase in vitality from the mixed-effects meta-analysis was 0.86 [95%CI 0.75 to 0.98], with modest to high heterogeneity across results ($I^2$ = 76%, p = 0.04) (**Fig 4**).

## Discussion

In this parallel analysis of real-world data from two nationally representative cohorts in Brazil and the United States, we consistently found higher patient-reported fatigue in the first 90 days after the start of PD was associated with a higher risk of all-cause mortality, even after adjustment for clinically important potential confounders. These results shed light on the complex dynamics between patient-reported and clinical outcomes among individuals living with kidney failure, particularly among cohorts of patients treated by the home kidney replacement therapy of PD in South and North America.

Multiple patient-reported outcomes metrics have been shown to associate with all-cause mortality, which has been consistently described for in-center HD patients, such as general HRQOL, functionality, fatigue, and post-dialysis recovery time [17]. On the other hand, there is a scarce understanding of patient-reported outcomes in the PD population, particularly during the incident period proximal to the initiation of the modality. In a previous analysis of the BRAZPD cohort in Brazil, lower baseline physical and mental composite scores were associated with a higher risk of all-cause mortality among incident PD patients [18]. Another prior study from the BRAZPD cohort showed functional status, as measured by the Karnofsky performance status, is a strong predictor of survival among PD patients [19]. A clinical trial posthoc study of Short-Form-36 data among prevalent and incident dialysis patients in Sweden also reported that lower physical composite scores were associated with higher risk of all-cause mortality in PD patients [20]. A clear definition of fatigue as the primary exposure, however, is lacking from the aforementioned studies.

**Table 2. Multivariate model estimates for all-cause mortality risk in the Brazil and United States cohorts.**

| Cohort | Brazil Cohort | | | United States Cohort | | |
|---|---|---|---|---|---|---|
| Model | Cox Proportional Hazards Model | Competing Risks | Multilevel | Cox Proportional Hazards Model | Competing Risks | Multilevel |
| | Estimate (95% CI) | Estimate (95% CI) | Estimate (95% CI) | Estimate (95% CI) | Estimate (95% CI) | Estimate (95% CI) |
| *Age > 65 years* | 2.53 (1.93 to 3.32) | 2.67 (2.03 to 3.51) | 2.77 (2.06 to 3.72) | 2.26 (1.97 to 2.6) | 2.26 (1.96 to 2.61) | 2.31 (2 to 2.66) |
| *Coronary artery disease* | 1.31 (0.98 to 1.75) | 1.28 (0.95 to 1.73) | 1.39 (1 to 1.92) | 1.3 (1.08 to 1.56) | 1.29 (1.07 to 1.55) | 1.33 (1.11 to 1.61) |
| *Diabetes* | 1.53 (1.17 to 2) | 1.49 (1.14 to 1.96) | 1.61 (1.2 to 2.17) | 1.41 (1.23 to 1.62) | 1.41 (1.22 to 1.62) | 1.43 (1.24 to 1.65) |
| *Initial Modality (CAPD)* | 0.77 (0.58 to 1.01) | 0.71 (0.54 to 0.94) | 0.76 (0.54 to 1.07) | 1.14 (1 to 1.3) | 1.16 (1.02 to 1.32) | 1.12 (0.97 to 1.29) |
| *Peripheral artery disease* | 1.55 (1.17 to 2.04) | 1.62 (1.23 to 2.15) | 1.64 (1.2 to 2.26) | 1.6 (1.27 to 2.01) | 1.69 (1.34 to 2.14) | 1.63 (1.29 to 2.06) |
| *Previous HD* | 1.32 (1.02 to 1.72) | 1.29 (0.99 to 1.68) | 1.4 (1.03 to 1.89) | 1.64 (1.43 to 1.89) | 1.61 (1.4 to 1.86) | 1.65 (1.42 to 1.9) |
| *Creatinine* | 0.97 (0.92 to 1.02) | 0.97 (0.92 to 1.02) | 0.95 (0.89 to 1) | 0.97 (0.95 to 0.99) | 0.97 (0.95 to 0.99) | 0.97 (0.95 to 0.99) |
| *Hemoglobin* | 0.87 (0.81 to 0.95) | 0.88 (0.82 to 0.95) | 0.87 (0.8 to 0.95) | 0.9 (0.86 to 0.94) | 0.88 (0.84 to 0.93) | 0.9 (0.86 to 0.94) |
| *Vitality Score* | | | | | | |
| *35 to <40* | 0.71 (0.47 to 1.05) | 0.75 (0.5 to 1.13) | 0.74 (0.47 to 1.14) | 1.04 (0.8 to 1.36) | 1.03 (0.79 to 1.33) | 1.04 (0.79 to 1.36) |
| *40 to <50* | 0.75 (0.55 to 1.02) | 0.8 (0.59 to 1.09) | 0.8 (0.57 to 1.13) | 0.78 (0.64 to 0.96) | 0.8 (0.65 to 0.99) | 0.76 (0.62 to 0.94) |
| *> = 50* | 0.5 (0.32 to 0.76) | 0.52 (0.34 to 0.8) | 0.47 (0.3 to 0.76) | 0.66 (0.57 to 0.77) | 0.67 (0.58 to 0.78) | 0.64 (0.55 to 0.75) |

Estimates for Cox proportional hazards and multilevel models present the hazard ratio and 95% confidence internals (CI) for all-cause mortality. Estimates for competing risks model present the sub-distribution hazard ratio and 95% confidence internals (CI) for all-cause mortality. CAPD: Continuous ambulatory peritoneal dialysis; HD: Hemodialysis.

Fatigue, defined as a complex multi-dimensional concept, encompassing both mental and physical domains, is highly valued by both HD and PD patients and their caregivers and providers [21]. In fact, among both PD individuals and caregivers, fatigue has been ranked to be as important as mortality in terms of priorities [5]. Although highly valued, fatigue has not yet been consistently explored in interventional studies as a core outcome for patients with kidney failure. Our results suggest that potential interventions that aim to improve the patients' reported symptoms at the incident dialysis period may improve clinical outcomes among PD individuals.

Fatigue has been described as one of the important barriers to home-therapy among ESRD individuals [4, 22]. Greater symptom burden during the incident dialysis period may adversely affect patient's and caregiver's perceptions of managing PD-related tasks, and therefore, can increase patient dependency, which is a common fear among kidney failure individuals [4]. Indeed, data from the Peritoneal Dialysis and Practice Patterns Study (PDOPPS) has shown patients who have a negative perception of PD modality tend to have lower HRQOL and are under a greater risk of technique failure over time [23]. We hypothesize fatigue symptoms at the transition phase to PD for kidney replacement therapy may disproportionally affect home-therapy patients, as a successful transition to dialysis requires greater patient engagement.

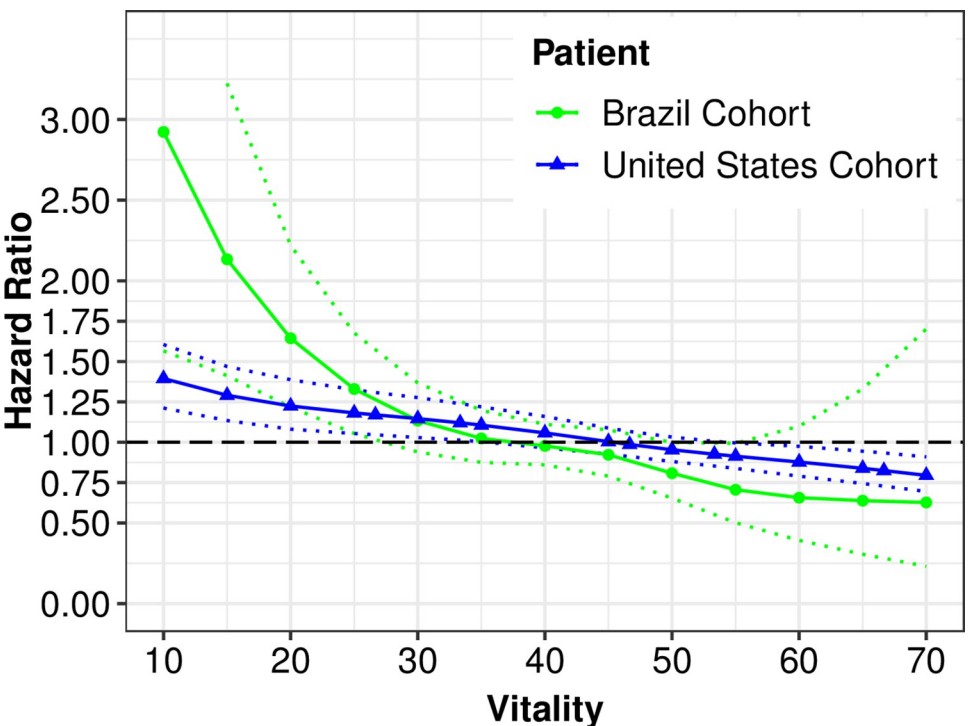

**Fig 3. Smoothing splines for the hazard ratio of mortality by continuous vitality scores in the cohorts from Brazil (Green) and the United States (Blue).** Models are adjusted for history of diabetes, coronary artery disease, peripheral artery disease, previous history of HD treatment, initial PD modality, hemoglobin, creatinine and age.

The results from this analysis suggest that improving patient symptoms, particularly at the late non-dialysis CKD stages, may positively affect post-dialysis outcomes, both for in-center and home-therapy dialysis. Indeed, under the framework of the continuum between early physical dysfunction in CKD to progressive sarcopenia and functional impairment, [24] exercise programs may promptly improve patients' symptoms, including fatigue, over time. A meta-analysis of randomized controlled trials investigating exercise programs among dialysis

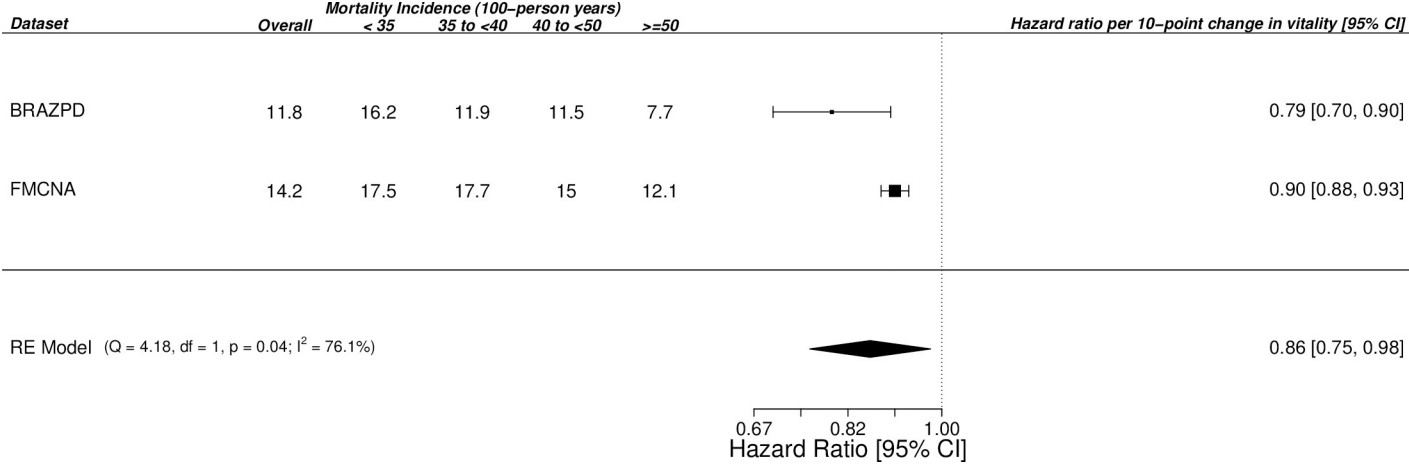

**Fig 4. Forest Plot for the random-effects meta-analysis of hazard ratios for each 10-unit increase in vitality scores.** CI: Confidence Interval.

and non-dialysis CKD patients has shown the intervention not only improves functionality but also results in better HRQOL and patient's symptoms [25]. Another randomized controlled trial, that included both in-center HD and PD patients, demonstrated a personalized home-based exercise program improves patients' cognitive and social function over 6-months. Therefore, physical rehabilitation may be one important strategy to improve the transition to home-dialysis therapy, a long overdue goal in terms of fostering a kidney replacement therapy that suits a person-centered model for ESRD care [26]. Whether such strategies for improving fatigue at the incident dialysis period improves mortality for PD patients is still unknown.

Our reported associations between vitality and all-cause mortality seem to be robust both at the modeling perspective, in which different models converge to consistent estimates and, more importantly, across a heterogeneous source of data. For this study, we included two nationwide representative cohorts for a parallel analysis and pooled estimates in a random effects meta-analysis. As expected, we found an important source of heterogeneity in our effect estimates, albeit both were consistent in directionality. This result is expected since there are potentially important sources of heterogeneity between cohorts. Notably, patients followed in the provider's cohort in the United States tended to have a shorter dialysis vintage as compared to those in the BRAZPD cohort in Brazil. This may justify the differences in terms of mortality rates between cohorts, which tended to be higher in the United States; this finding could be driven by the critical transition period from non-dialysis CKD to kidney replacement therapy that has been well described in previous studies, in which patients at earlier stages of transition to dialysis are under a higher risk of mortality [27].

Indeed, as graphically depicted in our smoothing spline regression plots, PD patients in Brazil that had the lowest vitality scores (highest level of fatigue) tended to have higher risks for all-cause mortality as compared to PD patients in the United States. Although merely hypothetical, these results might be driven by the subset of patients transitioning from HD to PD that could be more chronically burdened by fatigue; this is a more common subpopulation in Brazil as compared to the United States, and this is likely reflective of the distinct dialysis vintage between cohorts. In such subpopulation, higher fatigue burden at the transition to PD may disproportionally affect clinical outcomes, as lack of energy, comorbidity and inflammatory burden may affect PD technique training and execution. Further studies addressing the interactions between patient-reported outcomes, dialysis vintage, and kidney replacement therapy transitions are needed.

Our study has some limitations to note. Although we adjusted for important confounders, such as comorbidities, hemoglobin, and nutritional parameters, due to the observational nature of our study, we cannot rule out residual confounding. We included a select population that is only representative of patients who were treated with PD for greater than 90 days without a technique failure event, which may have resulted in a selection bias. Also, due to the study design, we included only patients who completed vitality scores within 90 days from PD start. This resulted in an exclusion of a relevant proportion of our sample. This may have resulted in selection bias because patients who completed vitality scores may be different from those who did it. Finally, we evaluated fatigue only at the period within 90 days from PD start. As fatigue is a dynamic exposure which may vary over time, a design evaluating longitudinal fatigue may be informative and should be considered in independent efforts.

However, our analysis has important strengths. We use two nationally representative cohorts in a parallel design that resulted in consistent results, which substantially increases the generalizability of our findings. In such approach, we leveraged distinct sources of data to increase generalizability, providing a framework of parallel analyses followed by a meta-analysis of random effects summarizing results. The inclusion of cohorts with potentially distinct practice patterns may further add to the generalizability, since our associations were consistent

in both datasets, despite the anticipated heterogeneities. This approach can be further employed in future initiatives in PD cohorts. Consistently, we explored different models to understand the associations between fatigue and mortality, which resulted in convergent estimates, regardless of distinct modeling assumptions.

## Conclusions

In summary, in this real-world multi-centric parallel analysis of nationally representative cohorts of incident PD patients in Brazil and the United States, we found that individuals who are more fatigued within 90 days of PD initiation had higher risks for all-cause mortality that was independent of key confounders. Strategies to improve patient-reported outcomes at dialysis initiation and their impact on clinical outcomes should be evaluated in intervention studies. Assessment of fatigue and PROs before transitioning to home dialysis may foster a better understanding of the patient and their potential barriers to care, which may yield a smoother person-centered transition to home dialysis, improve their quality of life, and increase survival.

## Supporting information

**S1 Table. Univariate model estimates for all-cause mortality risk in the Brazil and United States cohorts.**
(DOCX)

**S1 Fig. Kaplan meier curve stratified by vitality categories in the Brazil cohort.**
(PDF)

**S2 Fig. Kaplan meier curve stratified by vitality categories in the United States cohort.**
(PDF)

## Acknowledgments

The initial study results have been presented in an abstract at the 2021 World Congress of Nephrology [28]. We would like to acknowledge and thank the centers and providers represented in this side-by-side cohort analysis (BrazPD study participants and patients at the United States dialysis provider).

## Author Contributions

**Conceptualization:** Liz Wallim, Camila R. Guetter, Vladimir Rigodon, Chance Mysayphonh, Len A. Usvyat, Pasqual Barretti, Peter Kotanko, John W. Larkin, Franklin W. Maddux, Roberto Pecoits-Filho, Thyago Proenca de Moraes.

**Data curation:** Murilo Guedes, Yue Jiao, Len A. Usvyat, Thyago Proenca de Moraes.

**Formal analysis:** Murilo Guedes, Yue Jiao.

**Funding acquisition:** Franklin W. Maddux, Roberto Pecoits-Filho.

**Investigation:** Yue Jiao, Pasqual Barretti, Roberto Pecoits-Filho, Thyago Proenca de Moraes.

**Methodology:** Murilo Guedes, Liz Wallim, Yue Jiao, Len A. Usvyat, John W. Larkin, Roberto Pecoits-Filho, Thyago Proenca de Moraes.

**Project administration:** Murilo Guedes, Vladimir Rigodon, Len A. Usvyat, Peter Kotanko, John W. Larkin, Roberto Pecoits-Filho, Thyago Proenca de Moraes.

**Resources:** Pasqual Barretti, Roberto Pecoits-Filho, Thyago Proenca de Moraes.

**Software:** Murilo Guedes, Yue Jiao, Len A. Usvyat, Thyago Proenca de Moraes.

**Supervision:** Murilo Guedes, Len A. Usvyat, Pasqual Barretti, Peter Kotanko, John W. Larkin, Franklin W. Maddux, Roberto Pecoits-Filho, Thyago Proenca de Moraes.

**Validation:** Murilo Guedes, Yue Jiao.

**Visualization:** Murilo Guedes, Yue Jiao, John W. Larkin.

**Writing – original draft:** Murilo Guedes, Liz Wallim, Camila R. Guetter, Yue Jiao, John W. Larkin, Thyago Proenca de Moraes.

**Writing – review & editing:** Murilo Guedes, Liz Wallim, Camila R. Guetter, Yue Jiao, Vladimir Rigodon, Chance Mysayphonh, Len A. Usvyat, Pasqual Barretti, Peter Kotanko, John W. Larkin, Franklin W. Maddux, Roberto Pecoits-Filho, Thyago Proenca de Moraes.

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
