## [Decision Letter · Decision Letter 0]

15 Feb 2022

PONE-D-21-36672Fatigue in Incident Peritoneal Dialysis and Mortality: A Real-World Side-by-Side Study in Brazil and the United StatesPLOS ONE

Dear Dr. Rigodon,

Thank you for submitting your manuscript to PLOS ONE. After careful consideration, we feel that it has merit but does not fully meet PLOS ONE’s publication criteria as it currently stands. Therefore, we invite you to submit a revised version of the manuscript that addresses the points raised during the review process.

ACADEMIC EDITOR:

Interesting paper on a relevant topic. However, some concerns were raised by the reviewers (mostly regarding the justification of the study, the study population, Time period, and vitality scores) that should be addressed in a point-by-point fashion and revisions should be made accordingly. Please note that the revised MS will undergo peer review again and no guarantee can be given re: acceptance.

We look forward to receiving your revised manuscript.

Kind regards,

Frank JMF Dor, M.D., Ph.D., FEBS, FRCS

Academic Editor

PLOS ONE

Journal Requirements:

Upon re-submiting your revised manuscript, please upload your study’s minimal underlying data set as either Supporting Information files or to a stable, public repository and include the relevant URLs, DOIs, or accession numbers within your revised cover letter. For a list of acceptable repositories, please see http://journals.plos.org/plosone/s/data-availability#loc-recommended-repositories. Any potentially identifying patient information must be fully anonymized.

“I have read the journal's policy and the authors of this manuscript have the following competing interests: MG, LW, CRG, VMR, CM are students at Pontifícia Universidade Católica do Paraná. CRG is a student at Johns Hopkins Bloomberg School of Public Health. VMR, CM, YJ, JWL, LAU, FWM are employees of Fresenius Medical Care. PK is an employee of Renal Research Institute, a wholly owned subsidiary of Fresenius Medical Care. LAU, PK, FWM have share options/ownership in Fresenius Medical Care. JWL, LAU, PK, FWM are an inventor on patent(s) in the field of dialysis. PK receives honorarium from Up-To-Date and is on the Editorial Board of Blood Purification and Kidney and Blood Pressure Research. FWM has directorships in Fresenius Medical Care Management Board, Goldfinch Bio, and Vifor Fresenius Medical Care Renal Pharma. RPF, TPM are employed by Pontifícia Universidade Católica do Paraná, and are recipients of scholarships from the Brazilian Council for Research (CNPq). RPF is employed by Arbor Research Collaborative for Health, and receives research grants, consulting fees, and honoraria from AstraZeneca, Novo Nordisc, Akebia Therapeutics, and Fresenius Medical Care. JWL, PB, TPM are guest editors on the Editorial Board of Frontiers in Physiology.”

Reviewers' comments:

Reviewer's Responses to Questions

**Comments to the Author**

1. Is the manuscript technically sound, and do the data support the conclusions?

Reviewer #1: Partly

Reviewer #2: Yes

2. Has the statistical analysis been performed appropriately and rigorously? 

Reviewer #1: I Don't Know

Reviewer #2: Yes

3. Have the authors made all data underlying the findings in their manuscript fully available?

Reviewer #1: Yes

Reviewer #2: No

4. Is the manuscript presented in an intelligible fashion and written in standard English?

Reviewer #1: Yes

Reviewer #2: Yes

5. Review Comments to the Author

Reviewer #1: 1. The aim of this paper is to demonstrate that fatigue is related to mortality in the PD population. As this is true in the general population (fatigue being part of frailty), and has shown to be true in the HD population and as fatigue correlates with other markers of mortality, the authors should at least speculate why this may not be true for the PD population - and therefore that the study is worth carrying out. Was there ever any chance that this would not be true for the PD population - if not, is the study worth all the effort of carrying out?

2. Following on from above, why the 2 populations. They are very different.

3. Over what time period was the data collected?

4. It is stated that the observations are in incident patients - yet the BrazPD patients seem to have been on PD an average of >2years. So presumably BrazPD are prevalent and the US incident??

5. I am not a statistician

Reviewer #2: excellent paper - very interesting and well analysed

I wondered whethter might be a sensitivity analysis around patients who did not complete the vitality scores - ie were there outcomes broadly simlar to those who did. The reason is that the number who did complete the vitality score is a rather small number compared to the total group.

I note the numbers of events in the supplementary Kaplan Meier curves are rather small and wondered what the explanation for this is.

Do you have data on the repeatability of the vitality scores - it would be imporant to know

6. PLOS authors have the option to publish the peer review history of their article (what does this mean?). If published, this will include your full peer review and any attached files.

Reviewer #1: No

Reviewer #2: **Yes: **Martin Wilkie

---

## [Author Response · Author response to Decision Letter 0]

1 Apr 2022

Thank you for your thoughtful critiques. Please refer to the point by point response attached in the submission.

---

## [Decision Letter · Decision Letter 1]

7 Jun 2022

Fatigue in Incident Peritoneal Dialysis and Mortality: A Real-World Side-by-Side Study in Brazil and the United States

PONE-D-21-36672R1

Dear Dr. Rigodon,

We’re pleased to inform you that your manuscript has been judged scientifically suitable for publication and will be formally accepted for publication once it meets all outstanding technical requirements.

Kind regards,

Frank JMF Dor, M.D., Ph.D., FEBS, FRCS

Academic Editor

PLOS ONE

Additional Editor Comments (optional):

Reviewers' comments:

Reviewer's Responses to Questions

**Comments to the Author**

1. If the authors have adequately addressed your comments raised in a previous round of review and you feel that this manuscript is now acceptable for publication, you may indicate that here to bypass the “Comments to the Author” section, enter your conflict of interest statement in the “Confidential to Editor” section, and submit your "Accept" recommendation.

Reviewer #1: All comments have been addressed

2. Is the manuscript technically sound, and do the data support the conclusions?

Reviewer #1: Yes

3. Has the statistical analysis been performed appropriately and rigorously? 

Reviewer #1: Yes

4. Have the authors made all data underlying the findings in their manuscript fully available?

Reviewer #1: Yes

5. Is the manuscript presented in an intelligible fashion and written in standard English?

Reviewer #1: Yes

6. Review Comments to the Author

Reviewer #1: My concerns have been addressed. This is an interesting paper confirming that fatigue in incident PD patient relates to outcomes

7. PLOS authors have the option to publish the peer review history of their article (what does this mean?). If published, this will include your full peer review and any attached files.

Reviewer #1: No

---

## [Editor Report · Acceptance letter]

13 Jun 2022

PONE-D-21-36672R1 

Fatigue in Incident Peritoneal Dialysis and Mortality: A Real-World Side-by-Side Study in Brazil and the United States 

Dear Dr. Rigodon:

I'm pleased to inform you that your manuscript has been deemed suitable for publication in PLOS ONE. Congratulations! Your manuscript is now with our production department. 

Kind regards, 

on behalf of

Dr. Frank JMF Dor 

Academic Editor

PLOS ONE